# Evaluating insect-host interactions as a driver of species divergence in palm flower weevils

Bruno A. S. de Medeiros [1,2] & Brian D. Farrell[2]

Plants and their specialized flower visitors provide valuable insights into the evolutionary consequences of species interactions. In particular, antagonistic interactions between insects and plants have often been invoked as a major driver of diversification. Here we use a tropical community of palms and their specialized insect flower visitors to test whether antagonisms lead to higher population divergence. Interactions between palms and the insects visiting their flowers range from brood pollination to florivory and commensalism, with the latter being species that feed on decaying–and presumably undefended–plant tissues. We test the role of insect-host interactions in the early stages of diversification of nine species of beetles sharing host plants and geographical ranges by first delimiting cryptic species and then using models of genetic isolation by environment. The degree to which insect populations are structured by the genetic divergence of plant populations varies. A hierarchical model reveals that this variation is largely uncorrelated with the kind of interaction, showing that antagonistic interactions are not associated with higher genetic differentiation. Other aspects of host use that affect plant-associated insects regardless of the outcomes of their interactions, such as sensory biases, are likely more general drivers of insect population divergence.

[1] Smithsonian Tropical Research Institute, Panama City, Panama. [2] Museum of Comparative Zoology, Department of Organismic & Evolutionary Biology, Harvard University, Cambridge, MA, USA. ✉email: demedeirosb@si.edu

nsects comprise about two-thirds of the 1.5 million described species of animals[1], and current estimates predict that another 4 million insect species remain unknown[2]. This spectacular diversity is thought to be in a large degree a consequence of ecological speciation resulting from interactions with plants, particularly antagonistic interactions[3–10]. Antagonism between plants and insects could lead to accelerated rates of diversification, with the diversity of defenses among plants resulting from host specialization that in turn may spur radiations in insects circumventing those defenses[6,7,10–13]. The effects of these interactions are not restricted to macroevolution: theoretical models predict that, in specialized interactions, coevolution can lead to stronger differentiation when compared to spatial isolation alone in the case of antagonism but not in mutualism[14,15]. Regardless of the proximal mechanisms, a pattern of strong isolation by environment[16] may be expected when insect–plant interaction is a major cause of reduced gene flow and an insect species interacts with different plant species or populations. For example, in brood pollinators (specialized pollinators that are also seed predators[17]) it has been observed that more divergent host plant populations are associated with more divergent insect populations[18–23], but not in all cases evaluated[23,24].

If antagonisms promote divergent selection leading to the formation of host races and ecological speciation[9], genetic isolation between plant populations may be a better predictor of insect isolation in antagonists than in mutualists or commensals. It is unclear whether this is the case for most plant feeding insects, especially considering that these interactions often involve multiple partners and are spatially and temporally variable and context dependent[25]. Here we test this prediction by using a direct comparison between insects with different modes of interaction across scales of plant divergence. We take advantage of the variation in insect–plant interactions found in communities of palm-associated weevils distributed across the same geographic range and interacting with the same plants. We specifically test the hypothesis that isolation associated with host plant divergence is stronger in antagonistic species when compared to isolation by geographical distance alone.

Palms in the genus *Syagrus*, one of the closest relatives of the coconut[26,27], produce large inflorescences that are visited by dozens of insect species[28–32]. The most abundant flower visitors of these Neotropical palms are specialized beetles in the family Curculionidae, one of the most diverse insect taxa[33]. We recently described the community of insects associated with the seasonally dry forest palm *Syagrus coronata*, showing that many weevil species are broadly distributed throughout the plant geographical range[31]. Some of them are brood pollinators, while others are antagonists breeding on flowers or seeds and some are commensals breeding on decaying plant tissues. Populations of *S. coronata* have been found to have deep genetic divergences[34], and this plant shares many species of weevil with *Syagrus botryophora*, a parapatric palm specialized on rainforests and diverged

from *S. coronata* early in the history of the genus, about 20 million years ago[26,27]. Given this old divergence, weevil morphospecies shared by the two plants are likely a result of relatively recent host shifts as opposed to long-term co-diversification. We used double-digest RAD-seq (ddRAD), a low cost genome-wide sequencing method[35,36], to obtain genome-wide genetic markers for several populations of both plant species, including a population of *S. coronata* known as *S. × costae*, a hybrid with *S. cearensis*[26]. We used the same method to sequence nine morphospecies of weevil broadly distributed across the range of these palms. These nine morphospecies are all attracted to flowers and locally specialized on their host plants. They mate and lay eggs on their hosts and are distributed through a similar geographical range, but differ in the kind of interaction with plants in two relevant axes: their roles as pollinators as adults and whether their larvae breed on live or decaying tissues.

We first use the genomic data to delimit weevil species and better understand the diversity of these little-known insects. We find evidence for deeply divergent cryptic species, in most cases allopatric and associated with different hosts but broadly sympatric in the case of pollinators. Then, we test models of isolation by environment to ask whether the kind of interaction with host plants is associated with differences in the degree of isolation by geographical distance or isolation associated with host plant genetic divergence. Finally, we fit a Bayesian hierarchical model to test whether species with antagonistic interactions exhibit stronger levels of host-associated differentiation in relation to other species. We find that this is not the case: the variation in the degree of isolation by environment between species is not associated with breeding on live or decaying plant tissues.

## Results

**Cryptic weevil species.** Biological information on the species studied here is summarized in Table 1, and the geographic sampling in Supplementary Fig. 1. We initially assembled genomic data sets by filtering low-coverage loci (<12 reads) and genotyping each individual separately. Visualization of patterns of missing data revealed that, for some of the weevil species, certain ddRAD loci are shared within groups of samples, with very few loci recovered across groups (Supplementary Fig. 2). This pattern could be an artifact resulting from batch effects during ddRAD library preparation, because samples in a batch are pooled before size selection and PCR amplification[35]. Alternatively, it could be a consequence of cryptic, deeply differentiated taxa contained within each species as traditionally recognized by morphology[37]. Since studying the early stages of divergence does not make sense in the complete absence of gene flow[9,38], we first evaluated whether our data set included cryptic species.

To test whether this is the case, we recorded the number of loci shared, average sequence divergence, and batch identity for each pair of samples in each morphospecies. We found that samples processed in the same batch do share more loci, but extreme

**Table 1 Weevil morphospecies included in the study with references for natural history information.**

| Morphospecies | Pollinator | Larval breeding | Host palms |
|---|---|---|---|
| *Anchylorhynchus trapezicollis*[31,93] | Yes | Live tissue (developing fruits) | both |
| *Remertus rectinasus*[31] | No | Live tissue (developing fruits) | both |
| *Microstrates bondari*[94] | No | Live tissue (male flower buds) | both |
| *Microstrates ypsilon*[31] | No | Live tissue (male flower buds) | *S. coronata* |
| *Andranthobius bondari*[31] | No | Decaying tissue (aborted male flowers) | both |
| *Celetes impar*[31] | No | Decaying tissue (peduncular bract after anthesis) | *S. coronata* |
| *Celetes decolor*[31] | No | Decaying tissue (floral branches after fruit dispersal) | both |
| *Dialomia polyphaga*[31] | No | Decaying tissue (damaged inflorescences) | *S. coronata* |
| *Phytotribus cocoseae*[95,96] | No | Decaying tissue (peduncular bract after anthesis) | *S. botryophora* |

**Table 2 Effect of nucleotide distance and shared library batch on number of shared RAD loci (thousands).**

| Morphospecies | Intercept | Distance | Batch | $R^2$ |
|---|---|---|---|---|
| *Anchylorhynchus trapezicollis* | 7.4 (<0.01) | −1.5 (<0.01) | 0.3 (0.03) | 0.46 |
| *Andranthobius bondari* | 3.2 (<0.01) | −0.8 (<0.01) | 0.7 (<0.01) | 0.15 |
| *Celetes decolor* | 3.7 (<0.01) | −0.6 (<0.01) | 1.0 (0.03) | 0.15 |
| *Celetes impar* | 7.2 (0.36) | −0.4 (0.70) | 0.2 (0.71) | 0.002 |
| *Dialomia polyphaga* | 4.1 (0.11) | −2.9 (0.14) | 0.6 (0.03) | 0.09 |
| *Microstrates bondari* | 1.6 (0.87) | 0.4 (0.55) | 1.4 (0.01) | 0.08 |
| *Microstrates ypsilon* | 3.8 (0.29) | −0.7 (0.48) | 0.1 (0.70) | 0.01 |
| *Phytotribus cocoseae* | 18.5 (<0.01) | −24.2 (<0.01) | 1.4 (<0.01) | 0.21 |
| *Remertus rectinasus* | 3.4 (0.01) | −0.6 (0.01) | – | 0.04 |

All samples of *R. rectinasus* were prepared in the same batch. *P* values in parenthesis.

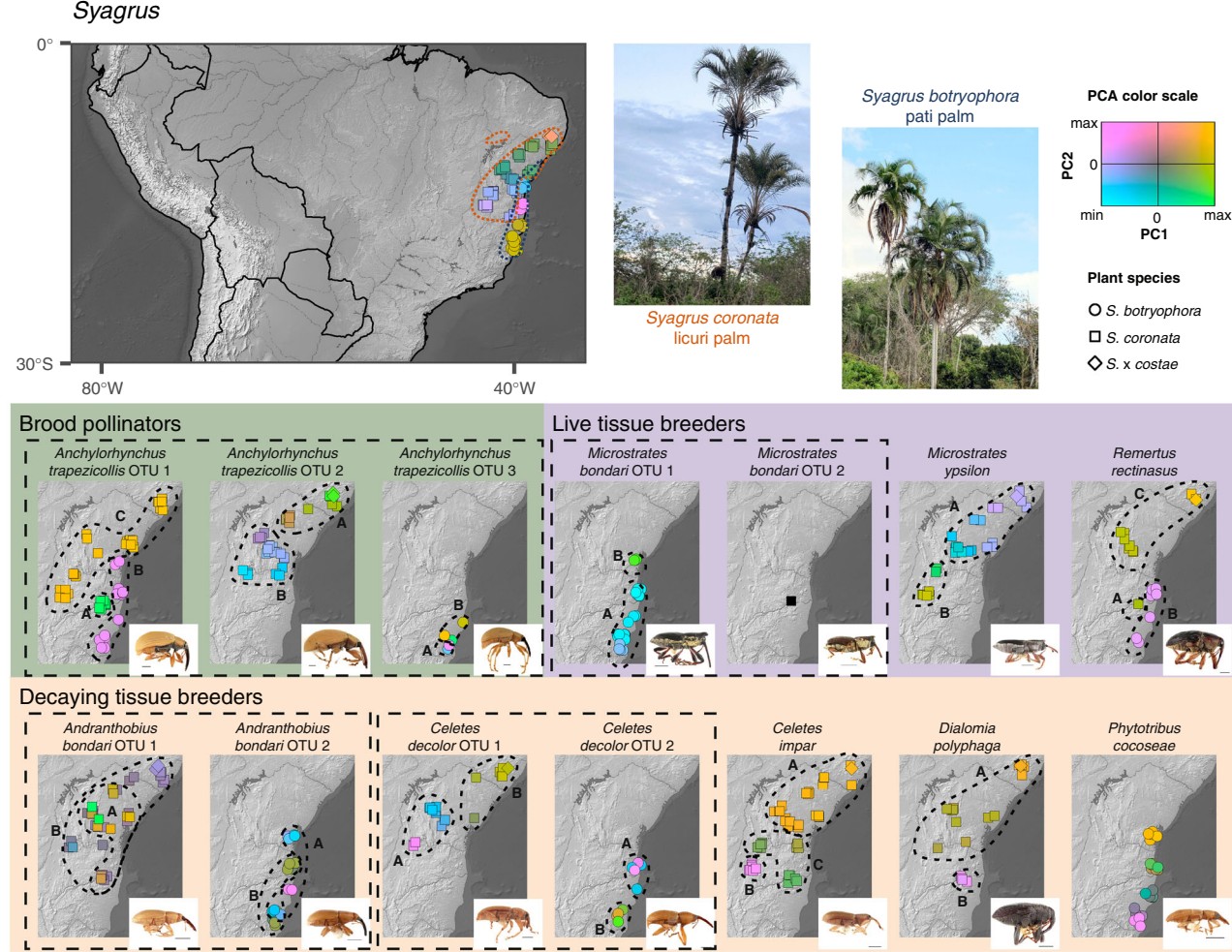

**Fig. 1 Principal component analysis (PCA) for each plant species and insect OTU.** The PCA for each plant and insect species is independent, with position of a sample in the first two PC axes coded following the color legend provided: samples with more similar colors have more similar PCA scores. *M. bondari* OTU 2 is black since no PCA is possible with a single sample. Supplementary Fig. 4 shows the same PCA results but plotted in traditional coordinates instead of colors in a map. A small jitter was added to enable visualization of overlapping points. Dashed boxes enclose morphospecies. Large map includes known palm distributions[26, 84] enclosed in dashed lines. Small maps show PCA results for each weevil OTU, with clusters enclosed in black dashed lines and labeled with uppercase letters corresponding to populations used in coalescent models (Table 3 and Supplementary Table 1). Scales 1 mm in insect images. Images of *A. bondari* OTU 1, *C. decolor* OTU 1, *C. impar*, *D. polyphaga*, *M. ypsilon*, and *R. rectinasus* OTU 1 were reproduced from de Medeiros et al.[31] with permission of Oxford University Press and The Linnean Society of London.

levels of missing data are only explained by deep sequence divergence, sometimes above 2.5% (Table 2 and Supplementary Fig. 2). We note that, in all cases, splitting samples into operational taxonomic units (OTU) at this level of sequence divergence results in groups with very high genetic differentiation from each other as measured by $G'_{ST}$ (Supplementary Fig. 2). With the exception of *Anchylorhynchus trapezicollis*, these clusters separate populations on each host plant (Fig. 1). For all

kinds of interactions, there is negligible to zero gene flow between these populations on the two different host plant species. In the case of the pollinator *Anchylorhynchus trapezicollis*, we find three genetic clusters, with one of them in both host species and broadly sympatric with the other two (Fig. 1). By comparing the morphology of the two most abundant clusters in sympatry and allopatry, we found differences in the length of ventral plumose hairs and in male secondary sexual characters (Supplementary Fig. 3). These diverged genetic clusters represent cryptic, previously unrecognized species. Hereafter, we will use OTUs as our unit of analysis, noting that these will be properly described as new species in the future. In general, we also recommend caution in studies of little-known organisms in which cryptic species might be common[39], noting that we were only able to distinguish OTUs because samples were individually barcoded and not pooled by location.

A principal component analysis (PCA) of the genetic variation of each OTU reveals little spatial congruence among weevil OTUs and variable congruence with the genetic variation of their host plants (Fig. 1). We found evidence for genetic clusters in 12 of the 13 weevil OTUs (Fig. 1 and Supplementary Fig. 4) and investigated whether there is gene flow between these clusters by using a model of isolation with migration based on the site frequency spectrum (Supplementary Fig. 5). We found that, in all cases, models including migration had higher support than those that did not (Table 3 and Supplementary Table 1). Populations of *Anchylorhynchus trapezicollis* OTU 1 and *Remertus rectinasus* on different host plants have much deeper divergence and smaller migration rates than those interacting with *S. coronata* alone (Table 3 and Supplementary Table 1), indicating that there are well-delimited host races even in these cases that divergence is shallow enough to enable assembly of ddRAD data sets.

**Interactions do not predict patterns of isolation.** Following evidence for ongoing gene flow between populations in each OTU, we assessed the role of geography and plant host as genetic barriers for each species of weevil. We also include climate in this analysis to account for the possibility of other differences in environment acting as genetic barriers. We used matrices of geographical distance, host plant genetic distance, and climatic distance between weevil populations as explanatory variables for the genetic covariance between weevils in a Bayesian model of isolation by distance and environment[40,41]. With model choice by cross validation, we found that climate was not a significant barrier to gene flow for any weevil species, and the significance of geography or host plant varied (Supplementary Table 2). For this

reason, we ran these models again using the full data set, but including only geography and host plant as predictors. The importance of geography or host plant as the main driver of divergence varied between weevil OTUs, and this variation seems uncorrelated to the mode of interaction (Fig. 2 and Supplementary Table 3).

To test whether species interactions are associated with differences in patterns of genetic divergence, we defined the statistics $\alpha_{diff}$, which describes the relative importance of host plants as sources of population divergence when compared to geography for a given OTU (see "Methods"). We then implemented a hierarchical Bayesian model to evaluate the independent effects of being a pollinator or breeding on live tissue (i.e., being an antagonist) on the value of $\alpha_{diff}$ (Eqs. (1) and (2)). We scored interactions along these two independent axes because the positive aspect of a brood pollination interaction may also affect rates of population divergence. Mutualisms have sometimes been claimed to lead to highly specialized interactions and thereby promote diversification in both insects and plants[42–44], but theoretical models do not predict that mutualisms lead to divergence in specialized interactions[15]. We note that *Anchylorhynchus* weevils are not exclusive pollinators of species of *Syagrus* palms[30,31,45] and the net effect of these interactions is currently unknown. Our model estimates the effect of pollination or antagonism by the parameters $\gamma_{pol}$ and $\gamma_{ant}$, respectively (Eq. (2)). A significantly positive value for these parameters means that pollinators ($\gamma_{pol}$) or antagonists ($\gamma_{ant}$) experience higher levels of divergence related to host plant divergence when compared to geography alone than species that are not pollinators or antagonists. We used posterior predictive simulations to find that the model adequately fits to the data (Supplementary Fig. 6), and also found that the number of OTUs used in this study provides enough power for inferences (see "Methods") (Supplementary Fig. 7). There is substantial variation in estimated $\alpha_{diff}$ between OTUs (Fig. 3a), but no evidence that $\gamma_{pol}$ is significant on either direction (Fig. 3b). While there is a positive trend for $\gamma_{ant}$, its 95% credibility interval includes negative values (Fig. 3b).

## Discussion

The degree of isolation by distance and by environment in these weevils co-distributed throughout the same range and interacting with the same plants varies widely, and this variation is largely unrelated to the kind of interaction with their hosts. All insect morphospecies previously thought to interact with both of two host plant species are actually comprised of cryptic species or

**Table 3 Summary of isolation-with-migration model fit, showing pairs of populations with direct gene flow inferred in the best and second best model, as well as the ∆AIC between them.**

| OTU | Populations included | Populations connected in best model | Populations connected in second best model | ∆AIC |
|---|---|---|---|---|
| *Anc. trapezicollis* OTU 1 | 3 | AB, AC, BC | AC, BC | 1848 |
| *Anc. trapezicollis* OTU 2 | 2 | AB | none | 5645 |
| *Anc. trapezicollis* OTU 3 | 2 | AB | none | 228 |
| *And. bondari* OTU 1 | 2 | AB | none | 6273 |
| *And. bondari* OTU 2 | 2 | AB | none | 2886 |
| *C. decolor* OTU 1 | 2 | AB | none | 3656 |
| *C. decolor* OTU 2 | 2 | AB | none | 182 |
| *C. impar* | 3 | AB, AC, BC | AB, BC | 1080 |
| *D. polyphaga* | 2 | AB | none | 902 |
| *M. bondari* OTU 1 | 2 | AB | none | 1096 |
| *M. ypsilon* | 2 | AB | none | 2620 |
| *R. rectinasus* | 3 | AB, AC | AB, AC, BC | 15 |

Population labels follow Fig. 1, full table with inferred parameters in Supplementary Table 1.

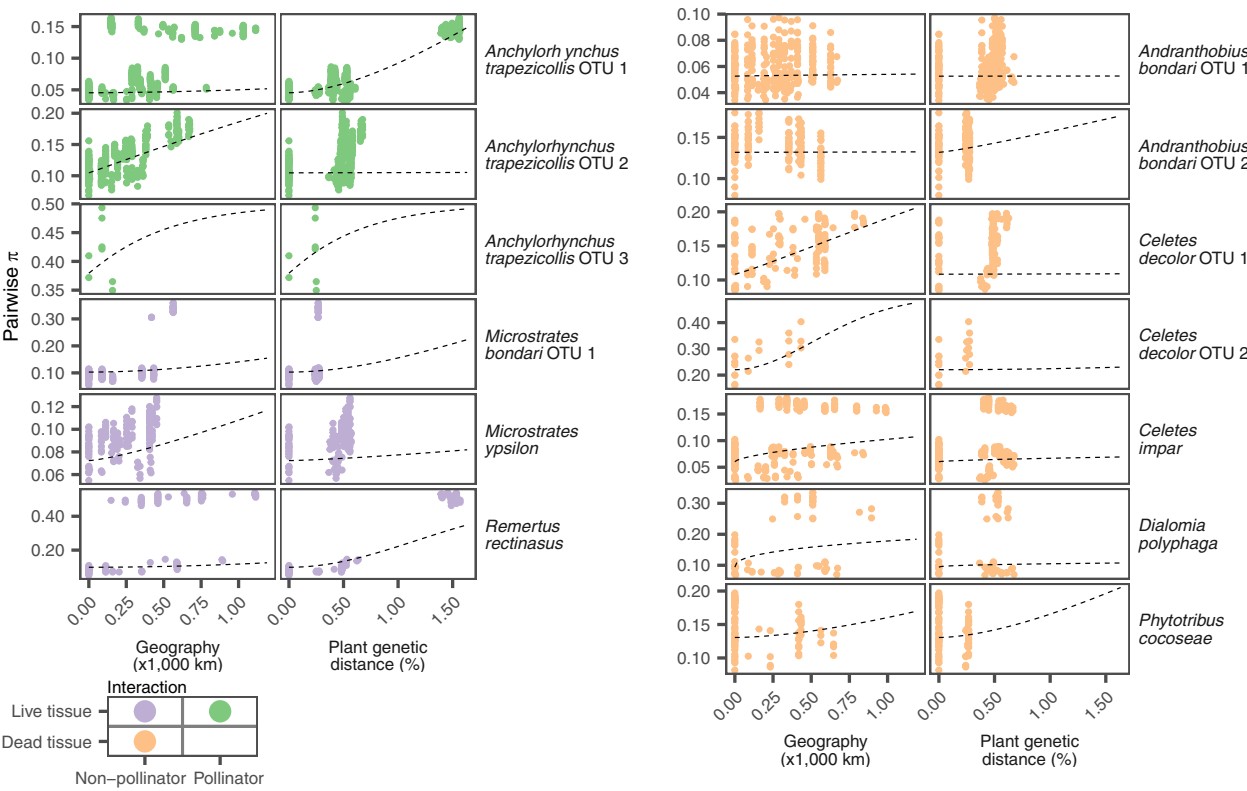

**Fig. 2 Effects of geographical and plant distance on weevil pairwise genetic distance in variable sites (pairwise $\pi$).** Colors show whether a species breeds on live tissue and is a pollinator following the color key. Dashed lines show the marginal effects of each distance implied by average parameter estimates.

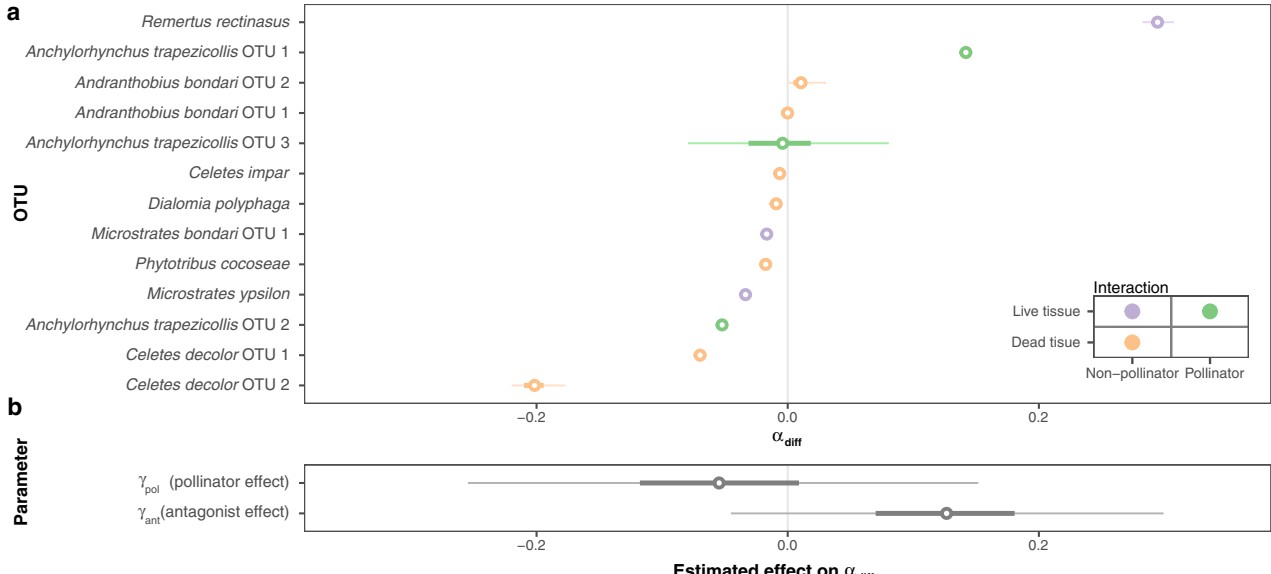

**Fig. 3 Effects of insect–plant interaction on variation of $\alpha_{diff}$ across OTUs. a** Posterior distribution of $\alpha_{diff}$ across OTUs, ordered by average $\alpha_{diff}$. **b** Posterior distribution of parameters associated with pollination and antagonism, when compared to non-pollinators and non-antagonists. Points: average estimates, thick lines: 50% credibility intervals, thin lines: 95% credibility intervals.

highly divergent populations, each specialized on a single host plant species. This is evidence that host plants constitute an important barrier for all beetle species sampled here. At a finer scale, plant host population divergence is a barrier to weevil gene flow for a subset of weevil OTUs, encompassing all kinds of interactions. OTUs breeding on live plant tissue seem to experience slightly higher divergence associated with host plants, but not significantly higher than other OTUs. Closely related OTUs do not necessarily show similar responses to geography and host plant divergence, suggesting that phylogenetically conserved traits (such as lifespan or flight ability) are not major drivers of the differences observed.

The lack of effect of pollination does not imply that mutualisms in general do not affect insect divergence rates. *Anchylorhynchus* weevils are not exclusive pollinators of *Syagrus* and therefore the outcomes of these interactions are very likely to be context-dependent and geographically variable, as other cases of non-specialized brood pollination[46]. Moreover, the difference in morphology of ventral hairs between OTUs might be related to pollen-carrying capacity. The lack of effect of antagonism, however, is unexpected. Palm flowers have chemical and physical defenses against herbivory[47], but weevil OTUs breeding on decaying tissues and therefore not interacting with these defenses exhibit similar patterns of isolation to those that do attack live defended tissues. While the ability to digest and detoxify plant tissues is thought to be a key adaptation enabling macroevolutionary diversification of phytophagous beetles[48], and weevils specifically[49], it is unlikely that coevolution and adaptation to plant defenses is a universal source of divergent selection and a necessary condition to explain the high rates of insect speciation. A recent review found that most studies on candidate genes for specialization to hosts in phytophagous insects focus on resistance or detoxification of plant secondary metabolites[50], but the actual source of selection might be in other aspects of host use. Divergence following host shifts is pervasive in phytophagous insects and their parasitoids[51], despite the large variation in interaction outcomes. Even though coevolution is an important driver of diversification under some conditions[5,7,15,43,52,53], evolution without reciprocal adaptation might be sufficient to explain many or most cases of insect specialization.

Diverse and complex phytophagous insect communities such as the one we study here are likely the norm rather than the exception in insect–plant interactions. Here we found that all weevil species, including those breeding on decaying plant tissues, show similar patterns of host-associated divergence. Strict antagonistic coevolution and divergence of host plant defenses are unlikely to drive this pattern. Despite the variation in larval breeding sites, all of the weevil species evaluated here mate on flowers[31], and it is possible that the usage of flowers as mating signals is a more general driver of divergence for these beetles and other phytophagous insects. Verbal models of how the evolution of sensory biases could be a major driver of phytophagous insects diversification have been proposed for a long time[54,55], but have received little attention in comparison to the wealth of research focused on plant defenses as drivers of diversity spurred by the classic study of Ehrlich and Raven[12]. The evolution of odorant receptors associated with mating signals in insect flower visitors has been recently linked to species divergence in at least one case[56]. Considering that about one-third of insect species visit flowers[57], the generality of flowers and other host plant cues working as mating signals that result in insect species divergence should receive more attention.

We studied patterns of isolation by distance and by environment in nine morphospecies of weevils associated with flowers of two palm species, which turned out to be 14 weevil OTUs after cryptic species were identified. Host plant species identity was a very strong barrier to gene flow in all cases, with a different OTU or a highly divergent population on each host. Both geography and host plants, but not climate, are important barriers determining genetic differentiation, with variation between insect species being largely unrelated to the kind of interaction with their host plants. Insect–plant antagonistic coevolution does not seem to be required for insect specialization and the generation of barriers to gene flow, and other aspects of insect–host interactions, such as sensory biases, should be investigated in studies of phytophagous insect diversification.

## Methods

**Sampling**. We sampled insects and plants from 13 populations of *S. coronata* (including *S. × costae*, hybrids with *S. cearensis*[26]) and five populations of *S. botryophora* throughout the distribution of both plant species (Fig. 1). Whole inflorescences were bagged and excised with insects aspirated and stored in 95% ethanol. Leaf tissues were collected from the sampled plant and other individuals in the vicinity. For this study, we chose nine specialized weevil species that we previously identified to engage into different kinds of interaction with their host plants (Table 1) and that have widespread geographical distributions[31] and sequenced one to ten individuals per morphospecies per locality (Supplementary Fig. 1).

**DNA extraction and library preparation**. We extracted DNA from insects and prepared ddRAD libraries[35] from 150 ng of input DNA as described in de Medeiros and Farrell[36], including whole-genome amplification for low-yield DNA extracts. Some of the individuals were extracted destructively, but for others we digested full bodies split at the base of the pronotum and preserved the remaining cuticle. For plants, DNA was extracted from leaf tissues using the E.Z.N.A. HP Plant DNA Mini Kit (Omega Biotek) following the manufacturer protocol, and libraries were prepared with the same enzymes and protocol as for insects, but from 300 to 1000 ng of genomic DNA without whole-genome amplification. Barcoded libraries were sequenced on Illumina systems, in several runs pooled with unrelated libraries. The minimum sequence length was single-end 100 bp, and all sequences were trimmed to this length prior to assembly.

**Initial data set assembly**. Sequences were demultiplexed by inline barcodes and assembled with ipyrad v.0.7.24[58,59]. For insects, sequences were entirely assembled de novo, but removing reads of potential endosymbionts by using the ipyrad option "denovo–reference" with reference sequences including genomes of known weevil symbionts[60] as well as *Rickettsia* and *Wolbachia* genomes downloaded from the NCBI. We assembled data sets separately for each insect morphospecies. For plants, sequences were assembled either by mapping to the draft genome assembly of the coconut[61] or de novo for unmapped reads, using the ipyrad option "denovo+reference". Reads were clustered within and between samples at 85% identity, and only loci with coverage greater or equal than 12 in a sample were retained for statistical base calling using ipyrad. Initially, we retained all samples and all loci present in at least four samples, and we used Matrix Condenser[36,62] to visualize patterns of missing data. We then removed samples with excessive missing data from the data sets, since with whole-genome amplification these are more likely to include contaminants and amplification artifacts[36]. Instead of choosing an arbitrary threshold for filtering, we flagged for removal outliers as observed in the histogram view of Matrix Condenser.

**Assessing missing data**. For each insect morphospecies, we calculated the following pairwise metrics: (1) number of loci sequenced in common for each pair of samples, (2) the average pairwise nucleotide distance using the function "dist.hamming" in R package phangorn v.2.4.0[63], and (3) whether the two samples were prepared in the same batch. We tested whether sequence distance and batch effects are negatively associated with the number of common loci by fitting a regression on distance matrix[64,65] implemented in the R package ecodist v.2.0.1[66].

**Assembly of final data sets**. After confirming that sequence distance is negatively associated with number of shared loci, we split the data sets for each morphospecies into clusters separated by at least 2.5% nucleotide differences using the R package dendextend v.1.8.0[67]. To further confirm if clusters thus obtained consist of highly isolated populations, we used the R packages mmod v.1.3.3[68] and adegenet v.2.1.1[69,70] to calculate $G'_{ST}$[71] between these clusters using all loci present in at least one individual per cluster. In the case of *Anchylorhynchus trapezicollis*, clusters were sympatric across a broad range, so we compared the morphology of individuals with preserved cuticle to confirm their divergence with an independent source of data. Sequencing statistics are available in Supplementary Table 4.

**Population structure**. We used bwa-mem v.0.7.15[72] to map reads on the consensus sequence for each RAD locus in the final data set. Alignment files in bam format were used as input to ANGSD v.0.920[73] and PCAngsd v.0.973[74] to filter sites not in Hardy–Weinberg equilibrium (HWE) while accounting for population structure[75]. We removed the whole RAD locus if any site was found not to be in HWE. We then used the same software to estimate genetic covariance matrices for each insect and plant species, as well as posterior genotype probabilities. PCA based on these covariance matrices were clustered by the k-means method with scripts modified from the R package adegenet. For each insect species, the optimal number of clusters was chosen by minimizing the Bayesian information criteria[76].

**Isolation with migration models**. We used ANGSD and dadi v.1.7.0[77] to generate the multidimensional site frequency spectra for each morphospecies with more than one k-mean cluster. We used these as input for models of isolation with migration[78] (Supplementary Fig. 5) in fastsimcoal v.2.6.0.3[79,80]. All simulations were done with a mutation rate of 3e−9, in line with other insects[81], but inferred parameters were finally scaled by the mutation rate (Supplementary Fig. 5).

For each model, we ran 100 independent searches of the maximum likelihood parameters and selected the best model by the Akaike information criterion (AIC).

**Isolation by distance and environment**. We used BEDASSLE v.2.0-a1[40,41] to infer the effects of geographical distance and host plant genetic distance on the genetic covariance of weevil populations. We additionally tested the effect of climatic distance as a confounder. We generated valid[82] (i.e., Euclidean) distances for explanatory variables as follows. We projected collection localities to UTM Zone 24S using the R package sf v.0.8-0[83] and calculated the Euclidean distance between them to obtain geographical distances. For climatic distance, we downloaded records of *S. coronata* and *S. botryophora* from GBIF[84] using the R package rgbif v.1.3.0[85], cleaned them with the R package CoordinateCleaner v.2.0-11[86], and then used the R package raster v.3.0-7[87] to extract bioclimatic variables[88] for these localities. We used PCA to find that the first PC explained 90.9% of the variance in the data set and that annual precipitation (bio12) had a very high loading on this component (Supplementary Fig. 8). Therefore, we used the difference in Annual Precipitation as climatic distance. For plant host genetic distances, we used NGSdist v.1.0.8[89] to estimate genetic distances between all samples of *Syagrus* based on posterior genotype probabilities and including invariant sites. We then calculated pairwise genetic distances between populations as the average distance between all of their samples, and checked that the resulting distances were Euclidean by using the R package ade4 v.1.7-15[90]. For each weevil OTU with three or more populations sampled, we called genotypes with posterior probability ≥ 0.8 and filtered the data set to one site per RAD locus to avoid linked sites, including only sites genotyped in at least one sample per population. For cross validation, we split data sets in ten partitions with 50 replicates and chose the simplest model among those with highest explanatory power. After finding that climate was not an important variable for any species, we ran BEDASSLE2 models on the full data set with only host plant and geography as distance matrices, with four chains of 2000 generations each and used the R package shinystan v.2.5.0[91] to evaluate convergence.

The BEDASSLE model estimates parameters associated with the strength of the relationship between a given distance matrix and the genetic isolation of species[40]. Here we denote $\alpha_g$ as parameter associated with geographical distance and $\alpha_p$ the parameter associated with host plant genetic distance. We used all samples from the posterior distribution to calculate $\alpha_{\text{diff}} = \alpha_p - \alpha_g$ for each OTU. The variation of $\alpha_{\text{diff}}$ between OTUs indicates the degree to which plant or geographical distances are associated with barriers to gene flow for each OTU, with more positive values associated with greater importance of host plants. We estimated the determinants of variation in $\alpha_{\text{diff}}$ across species by implementing a Bayesian hierarchical model similar to those typically used in meta-analyses. For each OTU $j$:

$$\alpha_{\text{diff}_j} \sim \text{Normal}\left(\theta_j, \sigma_j\right), \tag{1}$$

$$\theta_j \sim \text{Normal}\left(\mu + \gamma_{\text{ant}} \times I_{\text{ant}_j} + \gamma_{\text{pol}} \times I_{\text{pol}_j}, \tau\right). \tag{2}$$

In this model, $\sigma_j$ is the standard deviation in the posterior estimates for $\alpha_{\text{diff}}$, calculated from BEDASSLE posterior draws and assumed as known. $\mu$ is the mean $\alpha_{\text{diff}}$ for all OTUs, estimated by the model, and $\tau$ is the estimated variation in $\alpha_{\text{diff}}$ that is unrelated to species interactions. $I_{\text{ant}}$ and $I_{\text{pol}}$ are indicator variables for whether each species is an antagonist (i.e., breeds on live tissue) or pollinator, respectively (Table 1). In our data set, both indicators have a value of 1 for brood pollinators and 0 for non-pollinators breeding on dead tissue, while for non-pollinators breeding in live tissue $I_{\text{ant}} = 1$ and $I_{\text{pol}} = 0$. The parameters $\gamma_{\text{ant}}$ and $\gamma_{\text{pol}}$, therefore, are associated with the strength of the linear relationship between $I_{\text{ant}}$ and $I_{\text{pol}}$ and $\alpha_{\text{diff}}$, and constitute the model output of interest here. Values significantly different from 0 indicate that antagonism or pollination has a significant effect in determining the strength of weevil population divergence imposed by host divergence, when compared to space alone. We used standard Normal priors for $\gamma_{\text{ant}}$, $\gamma_{\text{pol}}$, and $\tau$ and $\mu$. We implemented this model in rstan v.19.2[92] using the Stan language. Models were run and convergence checked as for BEDASSLE models. We tested model fit by using posterior predictive simulations. Finally, we assessed whether the number of species included in this study is sufficient to achieve power to estimate $\gamma_{\text{ant}}$ and $\gamma_{\text{pol}}$ by running a model with an extreme case based on real data. We used the real distributions of $\alpha_{\text{diff}}$ but relabeled the three species with highest values as non-pollinator antagonists, the next three as both pollinators and antagonists, and the remaining seven as neither pollinators nor antagonists. This preserved the number of species for each category in the real data but maximized the differences in $\alpha_{\text{diff}}$ between modes of interaction.

**Statistics and reproducibility**. Sampling locations and sample sizes for all species are available in Supplementary Fig. 1. The number of samples, populations, and genetic markers for each OTU is available in Supplementary Table 4. When more than eight individuals for an insect morphospecies were available for sequencing in a locality, we arbitrarily chose eight individuals for DNA extraction. After discovering cryptic sympatric species in an initial analysis, we sequenced additional individuals targeting the putative species to confirm their identity. Moreover, we randomized the position of samples in DNA extraction plates to avoid potential biases arising from cross-contamination when performing high-throughput automated DNA extractions for insects[36]. Different statistical tests were used for each section of the manuscript, with details in the appropriate sections above.

**Reporting summary**. Further information on research design is available in the Nature Research Reporting Summary linked to this article.

## Data availability
Demultiplexed Illumina raw reads are deposited in the NCBI SRA, BioProject PRJNA397912, accessions SRR12602029–SRR12602364 for insect samples and SRR12603820–SRR12603892 for plant samples. Source data underlying graphs (Figs. 1–3 and Supplementary Figs. 1, 2, 4, 6–8) are available as Supplementary Data 1.

## Code availability
All R, Python, and bash scripts are available in the github repository https://github.com/brunoasm/rad_palm_weevil, with most R code provided as Rmarkdown notebooks including detailed annotation and visualization of intermediate steps.

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

## Acknowledgements

Sergio A. Vanin introduced the first author to entomology and to palm flower weevils, and assisted with weevil specimen handling and identification. Sergio Vanin passed away while this manuscript was in review, and this paper is dedicated to his memory. We thank Harri Lorenzi and Larry Noblick for palm location and identifications. Curators Renato de Melo Silva (SPF) and Sonia Casari (MZSP) allowed collection access and deposit of specimens, and Viviane Jono and Roberta Figueiredo helped with handling of plant specimens. Tauana Cunha, Elton Antunes, Filipe Gudin, Gabriel Pimenta, Zhengyang Wang, Luiz Fonseca, José I.L. Moura, Francisco José de Paula, and the Queiroz family assisted with fieldwork. Collections in Vale Natural Reserve, Córrego do Veado Biological Reserve, Descobrimento National Park, and Sooretama Biological Reserve were made with assistance from park authorities and permit from ICMBIO #39704-7. Work in the Museum of Comparative Zoology was assisted by Adrian Magaña, Scout Leonard, Dylan Ryals, Whit Farnum, and Amie Jones. Alyssa Hernandez produced SEMs. Harvard Bauer core staff helped with steps of library preparation and sequencing. Gideon Bradburd provided access to the development version of BEDASSLE2. The computations in this paper were run on the FASRC Cannon cluster supported by the FAS Division of Science Research Computing Group at Harvard University. The first author received a Jorge Paulo Lemann Fellowship for Research in Brazil. This work was funded by NSF DEB #1355169 and #1601356, the William F. Milton Fund (Harvard University), Putnam Expedition Grant (Museum of Comparative Zoology, Harvard University), and travel grants from the David Rockefeller Center for Latin American Studies. Publication was supported by a grant from the Wetmore Colles fund of the Museum of Comparative Zoology.

## Author contributions

B.A.S.M. designed the study, performed fieldwork, sorted and identified specimens, performed lab work, wrote computer code and the paper. B.D.F. contributed with the study design and writing the paper.

## Competing interests

The authors declare no competing interests.
