## [Peer Review File · Communications Biology]

Reviewers' comments:

Reviewer #1 (Remarks to the Author):

Evaluating species interactions as a driver of phytophagous insect divergence

Bruno A. S. de Medeiros and Brian D. Farrell tested in this paper the evolutionary consequences of insect-host interaction and its role in early stages of diversification in species of beetles that share host plants and geographic ranges. They used Palms in the genus *Syagrus*, one of the closest relatives of the coconut and its specialized beetles in the family Curculionidae, one of the most diverse insect taxa. They have tested different models of genetic isolation by environment, where they first delimit cryptic species. They show that insect populations varied and are structured by genetic divergence of plant populations. They show that this variation using a hierarchical model is not correlated with the kind of interaction. Moreover antagonistic interactions are not correlated with higher genetic differentiation. They conclude that the association regardless of their outcomes could be related to more general aspect such as sensory biases, which could drive insect's population divergence.

I have minor comments

1- Line 94 the authors used G_{st} for genetic differentiation. I would like to ask why they choose to use this statistics instead of other ones, such as F_{st} , D or even G'_{st} . We know that G_{st} is directly related to gene flow and mutation rate and the pattern is driven by migration when mutation rates are low in comparison to migration rates. If the mutation rate is very high, for instance, G'_{st} and D will be close to 1 and G_{st} will be close to zero for most markers and this is regardless of migration rates.

2- Figure 2 the font of the beetle species is kind of stretched.

3- Line 209, could they explain why they choose double-digest RAD-seq libraries method and not go with whole genome sequencing. Is it the cost of the sequencing or this is has to do with experimental design where they want to sequence more individuals/population?

4- Line 314 the RAD Illumina sequencing accession number is not available to access the data.

This study is well-designed and well-written paper with nice and easy language to understand. The method and supplement are nicely written and very informative. It is well excuted and tested a nice evolutionary hypothesis that would shed light and help understand insect-host interactions in other insect families and different hosts.

I recommend this paper for publication.

Reviewer #2 (Remarks to the Author):

Insects are one of the most diverse clades on Earth, and their evolutionary success is frequently attributed (at least in part) to the macroevolutionary consequences of mutualistic and antagonistic interactions with plants. However, the roles these interactions play in shaping microevolutionary processes, such as population differentiation, are less clear. This study seeks to understand how population differentiation is influenced by interactions that vary in their position along the mutualism-antagonism continuum. More specifically, they test whether antagonistic interactions with palm inflorescences lead to greater population differentiation in weevils. The study system included populations of two palm species, and nine weevil species (many representing multiple OTU's) that vary in their interaction type (antagonistic, commensalistic, mutualistic) in NE Brazil. The authors assembled an impressive molecular data set (ddRAD-seq for palms and over 400 weevil individuals) and used hierarchical modeling to reveal that population differentiation was not significantly higher in the three antagonistic weevil OTU's (relative to the three mutualists and seven commensalists). I was initially concerned about the sample size, but power analyses

confirmed that differences should be detected (if present), and provided further support for this non-significant result. I found this paper to be quite interesting and well-written, and it addresses a conceptual issue that will be of interest to a broad array of readers including those working on speciation, macroevolution, and biotic interactions. I have included some comments below that intended to further strengthen the manuscript.

Major Comments:

1. Throughout, it becomes confusing whether the units being discussed are weevil morphospecies or weevil OTU's. This is important to clarify, and it feels in many instances like they are being used somewhat interchangeably. I've highlighted a few cases in which "species" is used (which I initially interpreted as morphospecies), while I think OTU is the intended unit. I strongly suggest the authors carefully re-read and clarify this throughout.

L136: Were these conducted on OTU's or species? The first part of the sentence suggests OTU's, while the latter part implies species (Tables S2 and S3 suggest OTU's in the Tables themselves, but species in the legends).

L140, 143: Species? Or OTU?

2. I was also initially concerned about whether potential trends could be an artefact of phylogenetic relatedness (the three mutualists are from a single morphospecies in one genus, while the three antagonists are from two morphospecies in two separate genera). I found myself wondering what the phylogeny of these nine species (and even more OTU's) looked like, and whether this could be driving any patterns detected. As I read the paper, I could see that it was not driving the observed trends, but I think would be useful for the authors to articulate why this not the case (ie. the three closely-related mutualists actually have very different levels of population divergence instead of the similar levels that would be expected if phylogenetic relatedness was determining differentiation).

Minor Comments/Suggestions:

Figure 1: I am confused if separate PCA's were conducted for the insect covariance matrix as well as the palm covariance matrix. If separately, were PCA's done individually for each weevil morphospecies or OTU? Lines 120-122 clarify this (OTU), but I would recommend briefly mentioning in the Figure legend, and clarify further in the M&M (L254-257, which might suggest PCA's were done in species instead of OTU's).

L65: What is Comm Bio's policy on citing unpublished data/in prep?

L94-95: I'd suggest referring to Figure 1 and note which cluster doesn't separate by host plant.

L146: Mention that the two right columns (for dead tissue breeders) are again for geography and plant genetic distance (currently unlabeled).

Figure 3: Can OTU's (listed as species) in panel A be placed along the y axis? And would it make more sense to order OTU's in panel A by Interaction or Morphospecies?

Does Communications Biology require vouchers of individual plant and weevil species be deposited in herbaria/insect collections?

L234-235: Please clarify whether distance was calculated using the dist.ml function (and if so, which model?) or logdet, Hamming, polymorphism or Hadamard distance?

L240: Feels like a word is missing.

Reviewer #3 (Remarks to the Author):

The authors present genetic data of different weevil species to test if antagonistic interactions between insects and plants lead to greater diversification than mutualistic interactions. The authors conclude that there is great variation among species forming these different interactions in how much genetic distance is correlated with geographic or host plant distance, with no evidence that antagonistic interactions lead to greater diversification. Although I find this study system fascinating, I have some issues that should be addressed, especially my first comment below:

1. I am not convinced that this mutualistic interaction in particular should show lower genetic divergence associated with the host plant as initially expected. First, selection acting on the 'antagonists' might be similarly acting on the 'mutualists'. The brood is feeding on the plant's live tissue same as the other herbivores (classified as 'antagonistic'); thus, we could expect that selection to deal with the plant defenses (as stated in lines 178-179) should also be acting on the brood pollinators that also feed, at least as immatures, on the plant tissue. This is especially the case when 'antagonists' co-occur with 'mutualists', because if the plant evolves to deal with the antagonists, it will also have an effect on the mutualist.

Additionally, if the brood pollination mutualism is not obligate (and it seems that it is not at least for the plant), coevolution and diversification might occur differently than expected according to the model analyzed by Yoder et al. 2010 (table 1) for obligate brood pollination.

Second, the authors need to show that these pollinators in fact have a fitness advantage to the plant, because even if they are the most diverse visitors, the brood feeding could be more detrimental than the positive effect of pollinating. If that is the case, we would expect antagonistic coevolution (if there is coevolution at all in this system).

More detail about the system should be given to address these potential issues. I would also like to see a discussion on how the brood feeding on the plant could influence the pattern found for the interaction they defined as 'mutualistic'. It is hard to assess if diversification should in fact be stronger for the 'antagonists' without knowing if 1) mutualists and antagonists co-occur, 2) there is a net benefit of brood pollinators to the plant, 3) selection for matching of traits is stronger than selection against brood feeding.

2. L 34-35 : This sentence is not well connected with the rest of the paragraph, and later on (L43-45) there is conflicting information.

3. L 38-39: I believe the definition of brood pollinators specifically mention offspring feeding on seeds or reproductive tissue (following Hembry and Althoff 2016 Am J Bot).

4. L 57: please, include the reference for the recently described community.

5. Table 1: please, include the host plants used by these species.

6. L 96-98: the formation of cryptic species associated with the host plant might already be already of how host plant use can drive diversification in this system. I would like to see some discussion about this.

7. L 101-102: using morphological data to infer that a species is an exploiter is over-reaching. Is there evidence that these hairs in fact are used for pollen adherence? Is there evidence that the different type of hair in the species defined as exploiter leads to no pollen transfer?

8. It took me some time to understand that in Figure 1 the colors were relative to the location in the PCA (is that the right interpretation?). The figure is beautiful, I am wondering if there is a way to make it more intuitive... But other than that, I have a few specific questions/comments about the figure: What does the black square in the *Microstrates bondari* OUT 2 mean? Is that because there wasn't a PCA for this morphospecies as there is only one population? Also, here was the first time *Syagrus x costae* was mentioned, I think it should probably go into the section in the introduction where the authors talk about the system. Finally, to help interpret this figure, it would be helpful if figure S4 had similar shapes for the host plants.

9. L 122 – here it mentions that there was evidence of genetic clusters for only 6 of 13 weevil species, but in figures 1 and S4 and table 3 it seems that there are evidences for 12 out of 13.

10. L 136 – testing climate came out of nowhere here... the introduction should probably include what factors (other than host plant) could influence the isolation by environment (I am assuming this is the reason climate was included).

11. Analysis displayed in figure 3: why this analysis is only comparing pollinators and antagonists and not commensals? I see that in line 52 the authors specifically talk about comparing pollinators and antagonists, but in other parts of the paper they also compare commensals.

12. L 171 – when the authors talk about plant population divergence, it was only clear until I read the methods that they actually conducted a genetic analysis with the plants. Maybe in the introduction when talking about how host plants can drive diversification and what they did (e.g., 71-74), the authors could more explicitly say they used the genetic divergence of plants to infer isolation by host plant.

13. L 178-179: following my comment #1, it is hard to show if this is really the case when all the species tested were feeding on plant tissues.

14. L 187-188: the data don't necessarily show this, this is speculative.

15. L 192-193 – similar to my comment #6, this is already great evidence for diversification in response to host plants.

16. L 196-197: I don't think the data really show this, because pollinators were also feeding on the plant tissue. Even if the adults function mainly as pollinators, the offspring is still acting mainly as antagonists. And even if there is a net benefit of the pollinator to the plant when considering the effect of its brood, if there are other herbivores feeding on the same tissue, we could still find antagonistic coevolution acting on the pollinator. Also, is there evidence for coevolution in this system?

17. L 207: some of these have a really small sample size (1 or 2 individuals). Is this enough to infer population structure?

Our comments are marked in red.

We thank the three reviewers and the editor for the positive reviews and valuable suggestions for the manuscript. Below we include a response to each point raised by each reviewer. We would like to highlight that one of the reviewers made critical comments on the interpretation of beetle interactions considering that there is no data on fitness available. We believe that much of the apparent disagreement stems from a misunderstanding of how we used interaction data in our analysis, and we have therefore thoroughly reformatted the relevant sections in the manuscript and figures to make this clear. We agree that brood pollination, especially in the presence of other pollinators, may have a negative or positive net effect. We are not assuming that these interactions are mutualisms. What we do assume is that species breeding in live plant tissue may have negative effects on plant fitness and interact with plant defenses that evolved as a response to one or more of these species. This stands in contrast to species breeding on decaying plant tissue, such as aborted male flowers after anthesis or branches after fruit dispersal, which are not expected either to be important selective agents on plants or be affected by plant defenses. For this reason, we score each interaction along two axes: whether or not species are pollinators and whether or not species breed on live tissues. This was stated in the model formulation in the first version, but now we have changed color keys in figures and text in introduction and results to make this obvious. We hope that the new version of the manuscript and of the supplement can be acceptable for publication.

Referee expertise:

Referee #1:

Referee #2:

Referee #3:

Reviewers' comments:

Reviewer #1 (Remarks to the Author):

Evaluating species interactions as a driver of phytophagous insect divergence

Bruno A. S. de Medeiros and Brian D. Farrell tested in this paper the evolutionary consequences of insect-host interaction and its role in early stages of diversification in species of beetles that share host plants and geographic ranges. They used Palms in the genus *Syagrus*, one of the closest relatives of the coconut and its specialized beetles in the family Curculionidae, one of the most diverse insect taxa. They have tested different models of genetic isolation by environment, where they first delimit cryptic species. They show that insect populations varied and are structured by genetic divergence of plant populations. They show that this variation using a hierarchical model is not correlated with the kind of interaction. Moreover antagonistic interactions are not correlated with higher genetic differentiation. They conclude that the association regardless of their outcomes could be related to more general aspect such as sensory biases, which could drive insect's population divergence.

I have minor comments

1- Line 94 the authors used G_{st} for genetic differentiation. I would like to ask why they choose to use this statistics instead of other ones, such as F_{st} , D or even G'_{st} . We know that G_{st} is directly related to gene flow and mutation rate and the pattern is driven by migration when mutation rates are low in comparison to migration rates. If the mutation rate is very high, for instance, G'_{st} and D will be close to 1 and G_{st} will be close to zero for most markers and this is regardless of migration rates.

We used the corrected G_{st} of Hedrick (2005), also known as G'_{st} , but did not include a proper citation, leading to the confusion. We added the citation to the methods section in manuscript and replaced G_{st} with G'_{st} . [line 411]

2- Figure 2 the font of the beetle species is kind of stretched.

We reformulated the figure to reduce spacing in species labels. [Figure 2]

3- Line 209, could they explain why they choose double-digest RAD-seq libraries method and not go with whole genome sequencing. Is it the cost of the sequencing or this is has to do with experimental design where they want to sequence more individuals/population?

As pointed out by the reviewer, we wanted to maximize the number of individuals for the budget available. We added a sentence explaining that ddRAD is a low-cost genome-wide sequencing method. [line 83]

4- Line 314 the RAD Illumina sequencing accession number is not available to access the data.

We have now added SRA accession numbers and bioproject accession for this project. [line 524]

This study is well-designed and well-written paper with nice and easy language to understand. The method and supplement are nicely written and very informative. It is well excuted and tested a nice evolutionary hypothesis that would shed light and help understand insect-host interactions in other insect families and different hosts. I recommend this paper for publication.

We thank the reviewer for the kind comments.

Reviewer #2 (Remarks to the Author):

Insects are one of the most diverse clades on Earth, and their evolutionary success is frequently attributed (at least in part) to the macroevolutionary consequences of mutualistic and antagonistic interactions with plants. However, the roles these interactions play in shaping microevolutionary processes, such as population differentiation, are less clear. This study seeks to understand how population differentiation is influenced by interactions that vary in their position along the mutualism-antagonism continuum. More specifically, they test whether antagonistic interactions with palm inflorescences lead to greater population differentiation in weevils. The study system included populations of two palm species, and nine weevil species (many representing multiple OTU's) that vary in their interaction type (antagonistic, commensalistic, mutualistic) in NE Brazil. The authors assembled an impressive molecular data set (ddRAD-seq for palms and over 400 weevil individuals) and used hierarchical modeling to reveal that population differentiation was not significantly higher in the three antagonistic weevil OTU's (relative to the three mutualists and seven commensalists). I was initially concerned about the sample size, but power analyses confirmed that differences should be detected (if present), and provided further support for this non-significant result. I found this paper to be quite interesting and well-written, and it addresses a conceptual issue that will be of interest to a broad array of readers including those working on speciation, macroevolution, and biotic interactions. I have included some comments below that intended to further strengthen the manuscript.

We thank the reviewer for the kind comments.

Major Comments:

1. Throughout, it becomes confusing whether the units being discussed are weevil morphospecies or weevil OTU's. This is important to clarify, and it feels in many instances like they are being used somewhat interchangeably. I've highlighted a few cases in which "species" is used (which I initially interpreted as morphospecies), while I think OTU is the intended unit. I strongly suggest the authors carefully re-read and clarify this throughout.

Indeed after defining OTUs we started to use the term "species" to refer to these taxa, which might be confusing. We revised the text to refer to the taxa as recognized by morphology as "morphospecies", and to use OTU in all sections of the text after this term is defined, as well as in figure and table legends. [several lines throughout text and supplement]

L136: Were these conducted on OTU's or species? The first part of the sentence suggests OTU's, while the latter part implies species (Tables S2 and S3 suggest OTU's in the Tables themselves, but species in the legends).

We changed the text to clarify that these are OTUs [line 224]

L140, 143: Species? Or OTU?

We changed the text to clarify that these are OTUs [line 229, 232]

2. I was also initially concerned about whether potential trends could be an artefact of phylogenetic relatedness (the three mutualists are from a single morphospecies in one genus, while the three antagonists are from two morphospecies in two separate genera). I found myself wondering what the phylogeny of these nine species (and even more OTU's) looked like, and whether this could be driving any patterns detected. As I read the paper, I could see that it was not driving the observed trends, but I think would be useful for the authors to articulate why this not the case (ie. the three closely-related mutualists actually have very different levels of population divergence instead of the similar levels that would be expected if phylogenetic relatedness was determining differentiation).

We agree that it would be possible that phylogeny has a role. This could be observed, for example, if lifespan or dispersal distance were phylogenetically conserved. Since this information is not available currently, we believe that all we can say for now is that phylogenetically conserved traits are not major drivers of the patterns observed here, but they could play a role. We added one sentence to discussion on this topic [line 287]

Minor Comments/Suggestions:

Figure 1: I am confused if separate PCA's were conducted for the insect covariance matrix as well as the palm covariance matrix. If separately, were PCA's done individually for each weevil morphospecies or OTU? Lines 120-122 clarify this (OTU), but I would recommend briefly mentioning in the Figure legend, and clarify further in the M&M (L254-257, which might suggest PCA's were done in species instead of OTU's).

We clarified the usage of OTUs in this section. [lines 170-178, 433]

L65: What is Comm Bio's policy on citing unpublished data/in prep?

We removed reference to unpublished data, since it is not strictly necessary.[line 82]

L94-95: I'd suggest referring to Figure 1 and note which cluster doesn't separate by host plant.

We rephrased the sentence to note that the case is *A. trapezicollis*, and added reference to the figure. [lines 144-145]

L146: Mention that the two right columns (for dead tissue breeders) are again for geography and plant genetic distance (currently unlabeled).

We reformatted the figure and now all axis are labelled [Figure 2]

Figure 3: Can OTU's (listed as species) in panel A be placed along the y axis? And would it make more sense to order OTU's in panel A by Interaction or Morphospecies?

We added OTU names, but kept the ordering. The main reason for not showing names prominently and ordering by alpha_diff is to guide readers to focus on differences in the statistics, instead of the specific value for each species. This is because this shows results for the hierarchical model, with important relationships being between traits and alpha_diff, not species and alpha_diff. [Figure 3]

Does Communications Biology require vouchers of individual plant and weevil species be deposited in herbaria/insect collections?

Insect vouchers are deposited currently in the Museum of Zoology in São Paulo, on loan to Harvard University. These are in the process of being donated for final depositing at the Museum of Comparative Zoology, but this process will be very slow with the pandemic. Whenever it is completed, MCZ voucher ids will be linked to the SRA records, but this is not currently possible.

L234-235: Please clarify whether distance was calculated using the dist.ml function (and if so, which model?) or logdet, Hamming, polymorphism or Hadamard distance?

We added the function used (which implies the method): dist.hamming [line 405]

L240: Feels like a word is missing.

We rephrased the sentence. [line 410]

SI: Institution as Tropical

We corrected the typo in supplementary material.

Reviewer #3 (Remarks to the Author):

The authors present genetic data of different weevil species to test if antagonistic interactions between insects and plants lead to greater diversification than mutualistic interactions. The authors conclude that there is great variation among species forming these different interactions in how much genetic distance is correlated with geographic or host plant distance, with no evidence that antagonistic interactions lead to greater diversification. Although I find this study system fascinating, I have some issues that should be addressed, especially my first comment below:

1. I am not convinced that this mutualistic interaction in particular should show lower genetic divergence associated with the host plant as initially expected. First, selection acting on the 'antagonists' might be similarly acting on the 'mutualists'. The brood is feeding on the plant's live tissue same as the other herbivores (classified as 'antagonistic'); thus, we could expect that selection to deal with the plant defenses (as stated in lines 178-179) should also be acting

on the brood pollinators that also feed, at least as immatures, on the plant tissue. This is especially the case when 'antagonists' co-occur with 'mutualists', because if the plant evolves to deal with the antagonists, it will also have an effect on the mutualist.

Additionally, if the brood pollination mutualism is not obligate (and it seems that it is not at least for the plant), coevolution and diversification might occur differently than expected according to the model analyzed by Yoder et al. 2010 (table 1) for obligate brood pollination.

Second, the authors need to show that these pollinators in fact have a fitness advantage to the plant, because even if they are the most diverse visitors, the brood feeding could be more detrimental than the positive effect of pollinating. If that is the case, we would expect antagonistic coevolution (if there is coevolution at all in this system).

More detail about the system should be given to address these potential issues. I would also like to see a discussion on how the brood feeding on the plant could influence the pattern found for the interaction they defined as 'mutualistic'. It is hard to assess if diversification should in fact be stronger for the 'antagonists' without knowing if 1) mutualists and antagonists co-occur, 2) there is a net benefit of brood pollinators to the plant, 3) selection for matching of traits is stronger than selection against brood feeding.

We thank the reviewer for raising these important points, with which we generally agree. We rephrased several sentences throughout the text to make it more explicit that we are not equating pollination with mutualism, and that we are not ruling out the possibility of coevolutionary diversification in the cases in which interactions are extremely specialized (which is not the case here). Rather, our main goal was to test the effect of antagonisms, using breeding on live plant tissues as a proxy, but also accounting for the potentially mutualistic aspects of the interactions with brood pollinators, in a diverse community with many interacting species as is likely the case for most insect-plant interactions.

Specifically, we modified the main text in several sections and the color key in Figure 3 to make it clear that we decomposed interactions along two axes: whether each species breeds on live tissues as larvae and whether each species is a pollinator (regardless of effectiveness) as adult. The first axis is of main interest here and the focus of our discussion, but we consider that a model must include the second as well since it may be the case that pollinators have a mutualistic net effect. In the first version of the manuscript, we did not thoroughly discuss the lack of effect of pollination precisely because we agree this is not a very interesting result without knowledge on whether this represents true mutualism, and focused on the effect of mutualism. We now discuss this explicitly [line 302].

2. L 34-35 : This sentence is not well connected with the rest of the paragraph, and later on (L43-45) there is conflicting information.

The information is indeed conflicting, reflecting conflicts present in the literature (in this case, theoretical models vs empirical studies). We moved sentences and rephrased to make this more evident. [line 36]

3. L 38-39: I believe the definition of brood pollinators specifically mention offspring feeding on seeds or reproductive tissue (following Hembry and Althoff 2016 Am J Bot).

This is the narrower definition in Hembry and Althoff, but a broader definition including other tissues is also used in the literature (e. g. Sakai 2002, Journal of Plant Research). Since the narrow definition more closely aligns to our system of study, we updated the text and added the reference. [line 42]

4. L 57: please, include the reference for the recently described community.

The reference was provided in the sentence following this one, so we moved it for clarity. [line 77]

5. Table 1: please, include the host plants used by these species.

We included a new column with this information [Table 1, line 116]

6. L 96-98: the formation of cryptic species associated with the host plant might already be already of how host plant use can drive diversification in this system. I would like to see some discussion about this.

We agree. Maybe the discussion was obscured by the confusion with terms "species" and OTUs, but now, following reviewer 2 comments, we used a more standardized terminology. We also added a sentence to the discussion to make it more explicit. [line 283]

7. L 101-102: using morphological data to infer that a species is an exploiter is over-reaching. Is there evidence that these hairs in fact are used for pollen adherence? Is there evidence that the different type of hair in the species defined as exploiter leads to no pollen transfer?

This is indeed speculation and it was meant as a suggestion that this should be investigated and that not all OTUs could be equally good pollinators. Since the cryptic species were discovered in this study, follow-up work will be

required to evaluate this possibility. We removed it from this section to make it less distracting and added a note on a possible functional difference to the discussion without a specific reference to the evolution of exploiters in other systems. [line 305].

8. It took me some time to understand that in Figure 1 the colors were relative to the location in the PCA (is that the right interpretation?). The figure is beautiful, I am wondering if there is a way to make it more intuitive... But other than that, I have a few specific questions/comments about the figure: What does the black square in the *Microstrates bondari* OUT 2 mean? Is that because there wasn't a PCA for this morphospecies as there is only one population? Also, here was the first time *Syagrus x costae* was mentioned, I think it should probably go into the section in the introduction where the authors talk about the system. Finally, to help interpret this figure, it would be helpful if figure S4 had similar shapes for the host plants.

The interpretation is correct and we rewrote the figure legend to make it more clear, including a reference to Figure S4 showing the same data but without spatial information. We additionally reformatted the color key to make it more clear that it refers to PC axes 1 and 2. We also mentioned *S. x costae* in the introduction (it was mentioned in the methods section, but in the journal's format that is placed after results). Finally, we added symbols for host plants in figure S4 and the same population labels used in Figure 1, to enable more easy comparison between both figures. [Figure 2 in main text, Figure S4 in supplement]

9. L 122 – here it mentions that there was evidence of genetic clusters for only 6 of 13 weevil species, but in figures 1 and S4 and table 3 it seems that there are evidences for 12 out of 13.
Correct, it should be 12 out of 13.

10. L 136 – testing climate came out of nowhere here... the introduction should probably include what factors (other than host plant) could influence the isolation by environment (I am assuming this is the reason climate was included). We added a sentence to explain the rationale (which is correct, to account for the possibility that other factors are important). [line 224]

11. Analysis displayed in figure 3: why this analysis is only comparing pollinators and antagonists and not commensals? I see that in line 52 the authors specifically talk about comparing pollinators and antagonists, but in other parts of the paper they also compare commensals.
We have changed the wording of the methods section in which the model is explained, and also the figure color key to make it more clear that we are actually comparing two variables: whether or not a species is a pollinator, and whether or not a species is an antagonist. Therefore, we are comparing all species, including commensals. [color keys in Figure 2 and Figure 3, line 249, line 509]

12. L 171 – when the authors talk about plant population divergence, it was only clear until I read the methods that they actually conducted a genetic analysis with the plants. Maybe in the introduction when talking about how host plants can drive diversification and what they did (e.g., 71-74), the authors could more explicitly say they used the genetic divergence of plants to infer isolation by host plant.
We changed wording when describing sequencing [lines 83-86] and when describing plant distances [line 472].

13. L 178-179: following my comment #1, it is hard to show if this is really the case when all the species tested were feeding on plant tissues.
We respectfully disagree. A little over half of the species studied here breeds on decaying plant tissues after flowering or even fruit dispersal. There is no reason to believe they have strong effects on plant fitness or that plant defenses have a strong effect on their fitness. In contrast, the other species breed on male flower buds, before flowers open, or developing fruits. Table 1 presents only a summary of what is known on natural history to avoid the distraction of unnecessary details, but we include references to extensive work with details on these interactions. While the impact on plant fitness remains to be quantified, this is an important contrast: species that breed on live and potentially defended plant tissues against species that breed on dead plant tissue. If coevolution with plant defenses is a major factor leading to phytophagous insect population divergence, we would expect that the latter would not show as much divergence as the former, but we find that the two groups are indistinguishable in this regard, even when controlling for the fact that a subset of the live tissue breeders are also pollinators. We have rephrased the text to make these points more clear, added a sentence to discussion emphasizing this finding [line 306] and a paragraph with a more detailed discussion [starting in line 320]

14. L 187-188: the data don't necessarily show this, this is speculative.
This is a hypothesis that we believe is raised by our data and should be evaluated. We expanded this section and added more references to make this clear, starting in line 340.

15. L 192-193 – similar to my comment #6, this is already great evidence for diversification in response to host plants.

We agree, and this affects all species, regardless of how they interact with host plants. We hope that the several changes to discussion make it more clear now.

16. L 196-197: I don't think the data really show this, because pollinators were also feeding on the plant tissue. Even if the adults function mainly as pollinators, the offspring is still acting mainly as antagonists. And even if there is a net benefit of the pollinator to the plant when considering the effect of its brood, if there are other herbivores feeding on the same tissue, we could still find antagonistic coevolution acting on the pollinator. Also, is there evidence for coevolution in this system?

We believe the disagreement stems from a misunderstanding of how we used natural history information in our test: we did consider that brood pollinators were both pollinators and seed predators, because we scored the two traits separately. None of these traits showed evidence to have an effect on increasing or decreasing genetic divergence in relation to space. We hope that the new language in text and the new color keys in figures will help to clarify this. (see reviewer comment #11)

17. L 207: some of these have a really small sample size (1 or 2 individuals). Is this enough to infer population structure?

It is sufficient for genome-wide patterns, considering the very high number of markers.

For our specific analyses, clustering might suffer slightly in power if there are very closely related populations with only a few individuals, but we only use clustering to rule out the possibility that populations are completely isolated. A high degree of isolation would be found even with small sample sizes (for example, we were able to identify that the one sample of *Microstrates bondari* in a population of *Syagrus coronata* is actually a different species).

As for the model of isolation by environment, sample sizes that are too small would result in large posterior intervals, reflecting the prior variance of 1. For example, *A. trapezicollis* OTU 3, with only 5 individuals in 3 localities, has much larger posterior intervals than the other species. For most species, however, intervals are so short that they cannot even be visualized in Figure 3. Additionally, our hierarchical Bayesian model uses as input the means and intervals, fully accounting for any potential uncertainty due to small sample sizes.

REVIEWERS' COMMENTS:

Reviewer #2 (Remarks to the Author):

I previously served as a reviewer (R2) for this manuscript, and am pleased with the authors' revisions, which have addressed all of my queries.

Reviewer #3 (Remarks to the Author):

Medeiros and Farrell did a great job addressing my previous concerns. I believe that my problem with diversification rate comparison between mutualism and antagonism has been well addressed. I only have one minor comment that I believe wasn't exactly addressed from my previous review. Regarding figure 3, I like the changes made, but I think my comment wasn't very clear, I was specifically talking about part b of the figure. I still don't understand how weevils that feed on dead tissue were included in part b of the analysis. It seems that they are not being considered in this analysis, is that correct? Or if they are, they are included either with pollinators or with antagonists, and if that is the case, I don't think it would be the best approach as selection and divergence should be different for antagonists and these commensals. Why not have an additional measure for commensals only in part b of the analysis? The comparison including all 3 types of interactions would give a better idea of how antagonism or pollination would influence diversification in comparison with a more 'neutral' interaction.

Reviewer #2 (Remarks to the Author):

I previously served as a reviewer (R2) for this manuscript, and am pleased with the authors' revisions, which have addressed all of my queries.

Reviewer #3 (Remarks to the Author):

Medeiros and Farrell did a great job addressing my previous concerns. I believe that my problem with diversification rate comparison between mutualism and antagonism has been well addressed. I only have one minor comment that I believe wasn't exactly addressed from my previous review.

Regarding figure 3, I like the changes made, but I think my comment wasn't very clear, I was specifically talking about part b of the figure. I still don't understand how weevils that feed on dead tissue were included in part b of the analysis. It seems that they are not being considered in this analysis, is that correct? Or if they are, they are included either with pollinators or with antagonists, and if that is the case, I don't think it would be the best approach as selection and divergence should be different for antagonists and these commensals. Why not have an additional measure for commensals only in part b of the analysis? The comparison including all 3 types of interactions would give a better idea of how antagonism or pollination would influence diversification in comparison with a more 'neutral' interaction.

We thank the reviewer comments. Commensals are in fact included in fig 3b since they are the baseline comparison: y_{pol} is the effect of being a pollinator when compared to not being a pollinator and y_{ant} is the effect of being an atagonist when compared to not being an antagonist. We have changed the figure caption and axis labels in Fig. 3b to clarify this, which is already explained in detail in methods (lines 310-320).